# Mesitylated trityl radicals, a platform for doublet emission: symmetry breaking, charge-transfer states and conjugated polymers

Petri Murto [1], Rituparno Chowdhury[2,4], Sebastian Gorgon [2,4], Erjuan Guo [2,3,4], Weixuan Zeng [1,4], Biwen Li[2], Yuqi Sun [2], Haydn Francis[1], Richard H. Friend [2] ✉ & Hugo Bronstein [1,2] ✉

Neutral π-radicals have potential for use as light emitters in optoelectronic devices due to the absence of energetically low-lying non-emissive states. Here, we report a defect-free synthetic methodology via mesityl substitution at the *para*-positions of tris(2,4,6-trichlorophenyl)methyl radical. These materials reveal a number of novel optoelectronic properties. Firstly, mesityl substituted radicals show strongly enhanced photoluminescence arising from symmetry breaking in the excited state. Secondly, photoexcitation of thin films of 8 wt% radical in 4,4'-bis(carbazol-9-yl)-1,1'-biphenyl host matrix produces long lived (in the order of microseconds) intermolecular charge transfer states, following hole transfer to the host, that can show unexpectedly efficient red-shifted emission. Thirdly, covalent attachment of carbazole into the mesitylated radical gives very high photoluminescence yield of 93% in 4,4'-bis(carbazol-9-yl)-1,1'-biphenyl films and light-emitting diodes with maximum external quantum efficiency of 28% at a wavelength of 689 nm. Fourthly, a main-chain copolymer of the mesitylated radical and 9,9-dioctyl-9*H*-fluorene shows red-shifted emission beyond 800 nm.

Organic radical molecules are of special interest for luminescent applications because spin-allowed relaxation from their lowest-energy excited state, a doublet excited state ($D_1$), to the doublet ground state ($D_0$) provides a possibility to outperform organic closed-shell emitters which possess both singlet and triplet excited states[1,2]. They also benefit from fast emission process with nanosecond timescales, which reduces exciton quenching in electroluminescent applications. Majority of luminescent π-radicals are based on chlorinated triphenylmethyl (trityl) derivatives like tris(2,4,6-trichlorophenyl)methyl (TTM) and perchlorotriphenylmethyl (PTM)[3–6], and their pyridyl

nitrogen containing counterparts[7–9]. The reason is that the *o*-chlorine atoms provide superior steric protection stabilizing the carbon radical, whereas the other electron-withdrawing functionalities affect the frontier orbital energy levels (Supplementary Fig. 1).

Early studies predicted that the $D_1$ states of symmetric radicals would be dark[10,11]. Indeed, TTM and PTM are reported to have low photoluminescence quantum efficiencies (PLQEs) in the range of 1–3% in solution[8,12]. The poor emission of these alternant symmetry systems has been rationalized by degeneration of transition dipole moments arising from the highly symmetric molecular structure and relevant

[1]Yusuf Hamied Department of Chemistry, University of Cambridge, Cambridge CB2 1EW, UK. [2]Cavendish Laboratory, University of Cambridge, Cambridge CB3 OHE, UK. [3]Present address: State Key Laboratory of Materials Processing and Die and Mould Technology, School of Materials Science and Engineering, Huazhong University of Science and Technology, Wuhan, China. [4]These authors contributed equally: Rituparno Chowdhury, Sebastian Gorgon, Erjuan Guo, Weixuan Zeng. ✉e-mail: rhf10@cam.ac.uk; hab60@cam.ac.uk

molecular orbitals, that is, degenerate highest (doubly) occupied molecular orbital to singly occupied molecular orbital (HOMO–SOMO) and singly occupied molecular orbital to lowest unoccupied molecular orbital (SOMO–LUMO) electronic transitions[13,14]. To brighten the $D_1$ state, the key is to break the alternant symmetry of the radical[13,15]. Traditionally, this has been achieved by separating the individual electron and hole wavefunctions in the $D_1$ state with a non-alternant electron-donating group utilising the electron push-pull (D–A˙) design strategy. This can be understood as switching the doublet fluorescence from weak localized emission, typical for symmetric radicals like TTM and PTM, to emission with substantial charge-transfer (CT) character from derivatives like TTM-3PCz and TTM-3NCz (in the latter two structures TTM is covalently linked to the 3-position of 9-phenyl-9H-carbazole and 9-(naphthalen-2-yl)-9H-carbazole, respectively)[12,16]. The degeneracy of the lowest energy excitations is alleviated with a dominant HOMO (donor) to SOMO (radical) electronic transition, thus the HOMO energy level of the donor and the SOMO energy level of the radical define the optical energy gap (Supplementary Fig. 2). Non-alternant donors with a strong CT character have been utilised to make even symmetric D–A˙ radicals highly emissive[17]. The D–A˙ type radical design has ultimately led to the discovery of organic light-emitting diodes (OLEDs) with near-unity internal quantum efficiencies for emission peaking around 700 nm[16]. Further chlorination of the radicals, going from TTM to PTM, has improved their photostabilities by deepening the SOMO energies of the corresponding D–A˙ systems[18].

Apart from the above examples, only a handful of TTM derivatives have been reported for use in optoelectronic applications. Preparation of new and sophisticated radical structures is problematic and low-yielding because metal catalyzed reactions like Suzuki–Miyaura (S–M) cross-couplings lead to various chemical defects such as dehalogenation

of the p-chlorines[19]. Loss of o-chlorines is particularly detrimental to the stability of the radical[20,21]. Thus, despite their immense promise in optoelectronic devices, the lack of robust synthetic methodology is frustrating this class of materials in achieving their potential.

In this work, we present a synthetic methodology through which we have developed new families of luminescent π-radicals. These materials not only demonstrate superior performance but also allowed us to observe unexpected phenomena, opening the door to entirely new avenues of research in the field of luminescent radical materials. Importantly, substitution of TTM structure with bulky mesityl (M) groups, first introduced into trityl radicals by Hattori et al.[22], enables clean and selective synthesis of three families of radicals: (i) TTM with increasing level of mesitylation, (ii) sterically protected D–A˙ derivatives and (iii) main-chain conjugated polyradicals. Through study of these families, we report the following key advances: (1) symmetric (and alternant) TTM radicals, contrary to all previous reports, can show high luminescence, (2) TTM radicals can form highly emissive charge-transfer excitons with long-lived emission, (3) mesityl substitution also enhances the emission of D–A˙ type radicals and (4) conjugated poly-radicals can show efficient highly red-shifted photoluminescence (PL) in the near-infrared spectral region.

## Results and discussion
### Synthesis of three radical families
We introduce a series of singly, doubly and triply mesityl substituted TTM derivatives ($M_x$TTM, where x represents the number of mesityl groups) and demonstrate their versatile use in small molecular and polymeric radical synthesis. The three radical families synthesized here (i) the $M_x$TTM series, (ii) the D–A˙ type radical $M_2$TTM-3PCz and (iii) the polyradicals PFMTTM and PCzMTTM are shown in Fig. 1a and

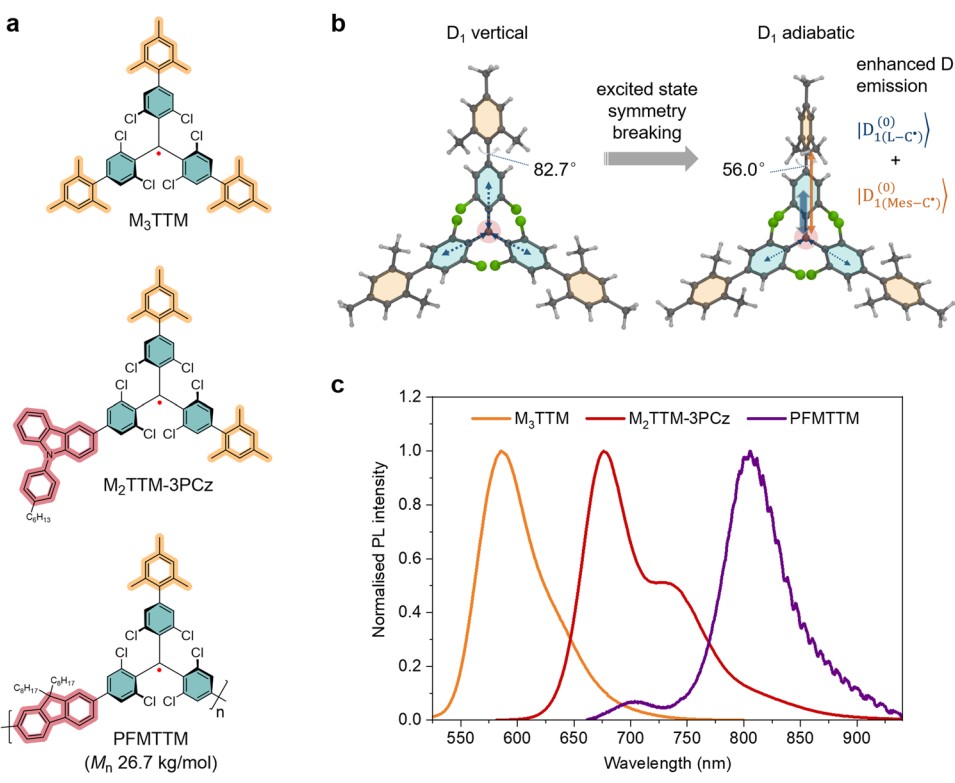

**Fig. 1 | Chemical structures of the radical families studied here and nature of their light emission. a** Chemical structures for $M_3$TTM radical and its conjugated D–A˙ radical and polyradical derivatives $M_2$TTM-3PCz and PFMTTM, respectively. The teal colour represents the TTM structure, whereas the orange and red high-lights stand for mesityl and donor groups, respectively. **b** Vertical and adiabatic excited states of $M_3$TTM radical based on its computationally optimized $D_0$ and $D_1$

geometries, respectively. Enhanced $D_1$ emission is described as short-range charge transfer contribution from both the chlorophenyl rings $|D_{1(L–C˙)}^{(0)}\rangle$ and the mesityl groups $|D_{1(Mes–C˙)}^{(0)}\rangle$. **c** Normalized PL spectra of $M_3$TTM, $M_2$TTM-3PCz and PFMTTM in 0.1 mM toluene solutions (the corresponding optical absorption spectra are included in Supplementary Fig. 5).

Supplementary Fig. 3 and discussed below. Detailed experimental protocols and systematic chemical names for all materials are included in Supplementary Note 1.

S–M coupling of αHTTM with mesitylboronic acid in mild anhydrous conditions gave the three precursor molecules αHM$_1$TTM, αHM$_2$TTM and αHM$_3$TTM in a single reaction without dehalogenation or other defects. As for further derivatives, it is generally expected that chlorine substituted aryl groups are less reactive in S–M couplings than bromine or iodine substituted ones due to the higher Ar–Cl bond dissociation energy[23,24]. To enhance the reactivity of non-mesitylated p-positions of αHM$_1$TTM and αHM$_2$TTM, their p-chlorines were converted to boronic acid pinacol ester (Bpin) groups via Miyaura borylation. This operation enabled selective S–M coupling with 3-bromo-9-(4-hexylphenyl)-9H-carbazole and clean synthesis of αH precursor of the corresponding mesitylated D–A˙ structure M$_2$TTM-3PCz.

The significance of mesityl substitution is clearly represented in S–M polycondensation of doubly Bpin functionalized αHM$_1$TTM molecule. Synthesis of αH precursors of PFMTTM and PCzMTTM was possible by coupling the mesityl-protected and Bpin-activated precursor with either 2,7-dibromo-9,9-dioctyl-9H-fluorene or 3,6-dibromo-9-(4-hexylphenyl)-9H-carbazole, ultimately yielding number-average molecular weights of 26.7 and 8.4 kg mol$^{-1}$ for the two polyradicals, respectively. The molecular weight of the former polymer is substantially higher than reported previously for TTM, PTM or any trityl radical based conjugated polymer system[19,25,26], whereas in the latter case lower molecular weight is expected for the 3,6-linked carbazole backbone[27–29]. Here, the mesityl substituents function as protecting groups effectively blocking reactions at the p-positions, whereas Bpin functionalization is essential in S–M polycondensations of radical precursors without which only low molecular weights and defected structures are obtained.

## Radical conversions and (reverse) α-hydrogenation reactions

Neutral π-radicals are NMR silent, but we demonstrate that their anionic equilibrium intermediates can be monitored quantitatively by NMR spectroscopy. The key to complete radical conversion is sufficient stabilization of the carbanion, which ultimately allows full synthetic control of the deprotonation–oxidation process. Figure 2a, b show the radical conversion of M$_x$TTM series and reaction monitoring by NMR spectroscopy (for details, see Supplementary Note 4). The black spectra represent the reaction mixtures at time zero (T0). Formation of metastable carbanions is observed as loss of αH signals upon addition of base. Despite stabilization in polar environment, the carbanion of TTM underwent spontaneous oxidation to neutral radical. Mesityl substituents provided added stability and a substantial fraction of M$_1$TTM was recorded in its anionic form, while M$_2$TTM and M$_3$TTM were totally stable in their carbanions, as monitored up to 3 h (see the emerging upfield shifted aromatic signals in Fig. 2b and Supplementary Figs. 14–17, teal spectra). Only the addition of p-chloranil initiated rapid oxidation to the corresponding M$_x$TTM radicals (Fig. 2b, red spectra). It is significant that mesityl substitution also stabilizes the anionic forms of conjugated D–A˙ radicals and polyradicals like those shown in Fig. 1a making the radical conversion procedure applicable in a wide range of TTM derivatives (see Supplementary Figs. 18–20).

To provide solid evidence of chemical integrity of the synthesized radicals, we reduced them back to the αH species with an excess of L-ascorbic acid as the antioxidant. α-Hydrogenation of the M$_x$TTM series is shown in Fig. 2a, c, while the D–A˙ radical and polyradicals are covered in Supplementary Note 4. In this case no anionic species was observed. Instead, the αH products (Fig. 2c, black spectra) emerged directly but slowly from the radicals (red spectra), suggesting that the carbanion is a transient species. In other words, reduction of the radical is the rate limiting step and hydrogenation of the carbanion is the fast step, which contradicts with the previously suggested reaction mechanism[30]. Another interesting observation is that mesityl

substitution stabilized the radical and slowed down the reaction further, so much so that α-hydrogenation of the M$_x$TTM series did not reach completion. NMR spectra of the αH products (Fig. 2c) confirmed that they were chemically same species as their αH precursors (Fig. 2b).

## Optical spectroscopy

We measured optical properties of these new materials in toluene solution (0.1 mM) and in solid films formed by vacuum sublimation (for M$_x$TTM series) or spin-coating (for M$_2$TTM-3PCz) with 4,4'-bis(carbazol-9-yl)-1,1'-biphenyl (CBP) at 8 wt% of the radical. Toluene provides an inert environment, whereas CBP acts as an electron donor, supporting intermolecular charge transfer. Optical properties are summarized in Table 1.

The optical absorption spectra of the M$_x$TTM series in toluene solution (Fig. 3a) show the typical high-energy D$_0 \rightarrow$ D$_m$ excitation band of TTM peaking in the UV spectral region, whereas the lowest-energy D$_0 \rightarrow$ D$_1$ excitation is centred around 545 nm (Table 1 and Supplementary Fig. 4). When going from TTM to the mesityl substituted radicals, a stronger feature is observed around 455 nm with the relative intensity increasing in the order TTM < M$_1$TTM < M$_2$TTM < M$_3$TTM. Excitation spectra of the series follow the same trend observed in their absorption spectra (Supplementary Fig. 33). Emission of TTM is weak in toluene solution, as expected from previous literature[13]. Mesityl substitution brings a systematic increase in PLQE by a factor of almost 20 for M$_3$TTM. Such PL enhancement is unexpected for M$_3$TTM, which could be expected to be non-emissive due to its symmetry. As developed below, this may arise from spontaneous excited-state symmetry breaking that is distinctively different from typical CT emission arising from D–A˙ hybrids in correlation between radiative decay rates and dihedral angles (see Supplementary Fig. 23 and the extended discussion in Supplementary Note 5)[31].

We note that mesityl substitution also enhances the PLQE of D–A˙ type radical M$_2$TTM-3PCz from 46% (for TTM-3PCz) to 84% in toluene solution, which stems from the increase in radiative decay rate (Table 1 and Supplementary Fig. 34)[16]. Moreover, PFMTTM and PCzMTTM stand as the first conjugated polyradicals that are not only emissive, but also feature a strong red-shift relative to the monomeric materials (emission max. >800 nm for PFMTTM) and high PLQEs in toluene solution (15 and 16%, respectively) for this wavelength range (Table 1, Fig. 1c and Supplementary Fig. 35). Their PLQEs are maintained in spin-coated thin films (13 and 10%, respectively) with no change in the respective peak positions (Supplementary Fig. 6).

We now turn to the emission of M$_x$TTM radicals in CBP thin films. With the exception of M$_3$TTM, the emission spectra of evaporated films of 8 wt% M$_x$TTM in CBP show significant red-shifting from their solution spectra (Fig. 3c). The partially mesitylated radicals have broadened and red-shifted emission, and the TTM film emission peaks beyond 700 nm. Remarkably, the emission for these CBP-host systems has near microsecond lifetimes (Fig. 3d), over two orders of magnitude slower than in toluene. We consider that the red-shifted and long-lived emission is from intermolecular charge transfer states with CBP acting as electron donor, in agreement with previous computational studies of Abroshan, et al.[31,32] on radical emitter:CBP host clusters. The red-shift is independent of doping concentration and specifically observed in CBP films (Supplementary Fig. 7), excluding the possibility of formation of long-lived intermolecular excimers (Supplementary Fig. 36 and 37)[33,34]. Thus these charge transfer states are intermolecular analogues of the highly emissive intramolecular states with CT character present in D–A˙ radicals such as TTM-3PCz, TTM-3NCz and TTM-1Cz[13,16,35–37]. Despite these slow emission rates, the PLQEs remain uncommonly high, up to 28% for M$_3$TTM (Table 1), indicating that nonradiative decay channels are significantly suppressed. The emission in CBP films of the two partially mesitylated radicals red-shifts in time (Fig. 3f). Emission from the M$_1$TTM film has a peak at 650 nm at early times, but it rapidly red-shifts beyond 700 nm on a timescale of

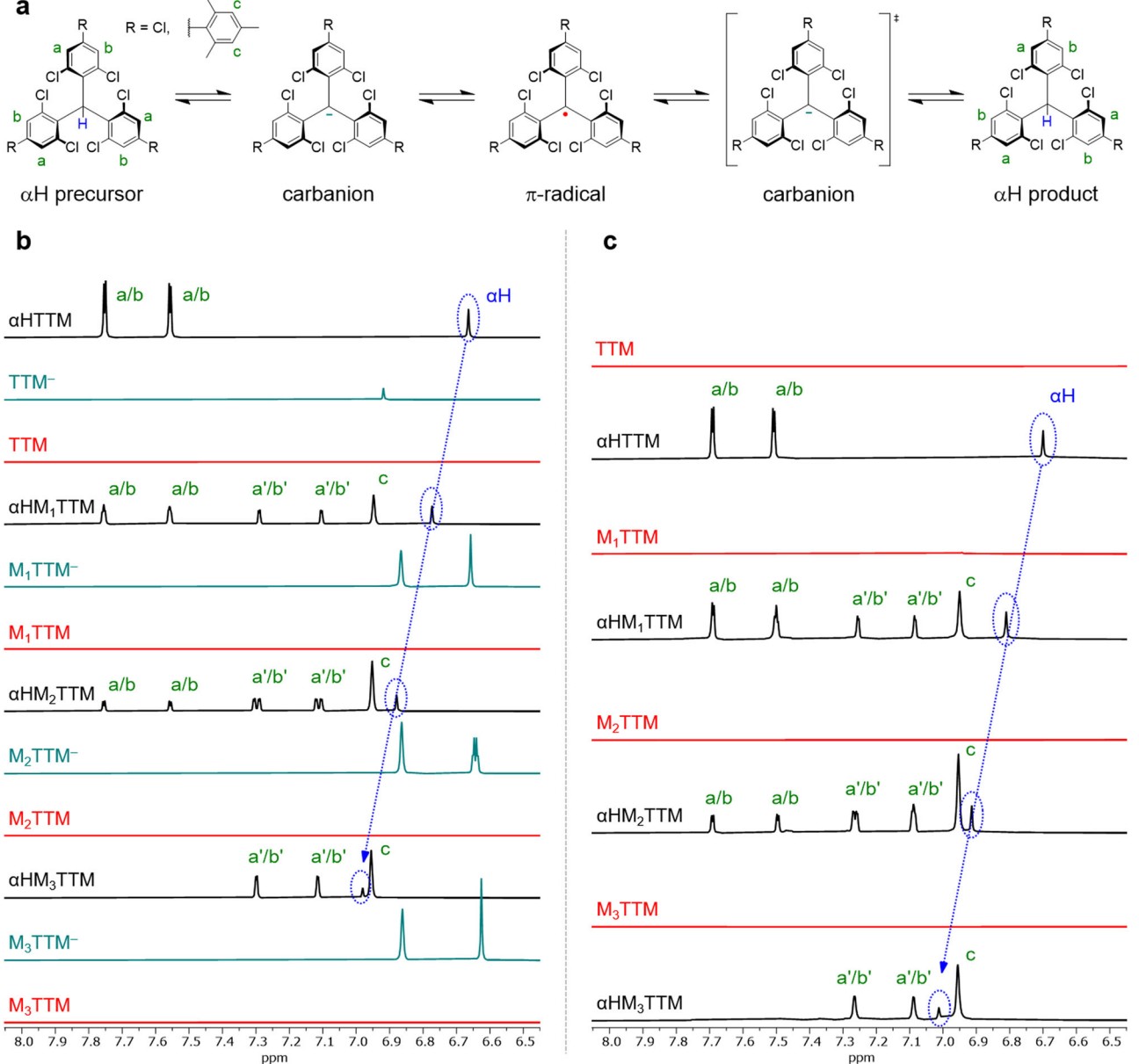

**Fig. 2 | Radical conversion and α-hydrogenation of the $M_xTTM$ series.**
**a** Schematic illustration of the reaction steps involved in radical conversion followed by α-hydrogenation. **b** Deprotonation of the αH precursor followed by one-electron oxidation in DMSO-$d_6$/THF-$d_8$ 3:1 (v/v) mixture. Partial $^1$H NMR spectra of the αH precursors (black lines), carbanions (teal lines) and radicals (red lines). **c** α-Hydrogenation of the radical in DMSO-$d_6$/THF-$d_8$ 1:1 (v/v) mixture. Partial $^1$H NMR spectra of the radicals (red lines) and their α-hydrogenated products (black lines).

No carbanion species was observed in the reverse reaction. The green letters indicate aromatic protons and their corresponding $^1$H signals. Mesityl substituted chlorophenyl ring protons are distinguished from the non-mesitylated ones by their upfield shifts, as indicated by the apostrophes. The blue arrows illustrate a systematic downfield shift of the αH signal along with increasing mesityl substitution. All spectra are referenced against 1,3,5-trimethoxybenzene ($^1$H, 6.09 ppm) as the internal standard.

tens of ns, explaining the steady state peak at 687 nm. In $M_2TTM$ film the red-shift is slower (Fig. 3e). The inset to Fig. 3d shows early time decay kinetics; the cross-over from $M_xTTM$ molecular emission to delayed emission occurs within the first 20 ns, fastest for TTM and slowest for $M_3TTM$. We attribute the broader, red-shifted emission that develops at later times to emission from charge-transfer excitons. However, we still observe $M_xTTM$ molecular emission at long times, particularly for $M_3TTM$. We propose that there is competition between direct intermolecular CT emission and endothermic regeneration of the molecular exciton, with an energy barrier of $59 \pm 13$ meV estimated for $M_3TTM$ (Supplementary Fig. 38). These long recombination times are independent of excitation fluence, in the range 5 to 40 μJ cm$^{-2}$ (Supplementary Fig. 39) and indicate that CT excitons remain bound.

We performed transient luminescence spectroscopy over the temperature range 295 to 6 K for the CBP-hosted films. The emission rate is unchanged for TTM but becomes slower at lower temperatures for both $M_2TTM$ and $M_3TTM$ (Supplementary Fig. 40); the time to 50% integrated emission is delayed from 400 ns at room temperature to 6 μs at 6 K for $M_3TTM$. TTM films show no spectral evolution at 295 or 6 K, with emission centred near 700 nm throughout their decay (Supplementary Fig. 41). The red-shift of emission as a function of time present in $M_2TTM$ at room temperature is faster and larger upon cooling to 6 K (Supplementary Fig. 42), indicating that the step reforming the molecular exciton is frozen out. Similarly, while $M_3TTM$ displays no emission red-shift at room temperature, the spectral evolution at 6 K is analogous to the 295 K dynamics in $M_2TTM$ (Supplementary Fig. 43).

**Table 1 | Photophysical parameters of the synthesized radicals**

| Radical | $\lambda_{abs}$, soln (nm)[a,b] | $\lambda_{PL}$, soln (nm)[a,c] | $\Phi_{PL}$, soln (%)[a,d] | $\tau_{PL}$, soln (ns)[a,e] | $k_r$, soln (×10⁶ s⁻¹)[a,f] | $k_{nr}$, soln (×10⁶ s⁻¹)[a,g] | $\lambda_{PL}$, film (nm)[c] | $\Phi_{PL}$, film (%)[d] |
|---|---|---|---|---|---|---|---|---|
| TTM | 375, 546 | 568 | 1 | 6.7 | 1.8 | 147.5 | 708[h] | 8[h] |
| M₁TTM | 374, 453, 545 | 593 | 2 | 14.2 | 1.5 | 68.9 | 687[h] | 13[h] |
| M₂TTM | 373, 453, 546 | 593 | 11 | 22.6 | 4.8 | 39.4 | 628[h] | 22[h] |
| M₃TTM | 372, 454, 537 | 586 | 23 | 28.1 | 8.2 | 27.4 | 594[h] | 28[h] |
| M₂TTM-3PCz | 379, 590 | 676 | 84 | 7.8 | 107.9 | 20.3 | 710[i] | 93[i] |
| PFMTTM | 462, 591 | 805 | 15 | 6.8 | 22.2 | 124.9 | 808[j] | 13[j] |
| PCzMTTM | 383, 638 | 683 | 16 | 7.5 | 21.7 | 111.6 | 687[j] | 10[j] |

[a]Sample in 0.1 mM toluene solution.
[b]Peaks of UV-vis absorption.
[c]Photoluminescence peak wavelength.
[d]Photoluminescence quantum efficiency.
[e]Photoluminescence lifetime.
[f]Radiative decay rate.
[g]Nonradiative decay rate.
[h]Evaporated film of 8 wt% radical in CBP matrix.
[i]Spin-coated film from toluene solution, 8 wt% radical in CBP matrix.
[j]Spin-coated film from toluene solution.

This is the first observation of intermolecular charge-transfer state emission in TTM radicals, and we show its kinetics can be controlled both thermodynamically and sterically by mesitylation of the TTM moiety. Thus, the contribution of the CT emission channel can be tuned, with TTM showing pure CT emission, M₁TTM and M₂TTM showing mixed molecular and CT emission, and only the fully protected M₃TTM showing pure molecular emission at room temperature.

## Quantum-chemical modelling

We have performed atomic dipole moment corrected Hirshfeld (ADCH) charge analysis in both D₀ state and adiabatic D₁ state geometries (Supplementary Figs. 24–27) for the MₓTTM series. In the D₀ → D₁ excitation, an electron moves from the chlorophenyl rings (here, referred to as ligands, L) to the central carbon radical indicating a ligand-to-centre CT transition for the excitation with a charge difference of about 0.07 e at the central carbon. The D₁ → D₀ emission goes reversibly as a centre-to-ligand CT transition. Correlation of the electronic transitions with the excitation and emission processes is further confirmed by the hole-electron analysis of the relevant excited states (Supplementary Note 5). The CT distance is rather small, which suggests that the absorption and emission spectra feature fine structures typical for a local excitation. Interestingly, the mesityl substituents contribute minimally to the electron spin density distribution as evidenced by the small charge differences (0.005 e) during the electronic transitions between D₀ and D₁ states suggesting that the increased emissivity of these materials is predominantly a non-electronic, activating effect by the mesityl groups.

We note that in the relaxed D₁ geometries, the short-range centre-to-ligand CT state tends to shrink and localize on one of the ligands with slightly increased dihedral between the ligand and the radical centre, while the dihedrals of the other two ligands decrease (Supplementary Table 4). This indicates spontaneous excited-state symmetry breaking is occurring, as the point group of TTM changes from *D*3 in the D₀ state to *C*2 in the D₁ state rendering it slightly emissive (~1%). The same effect operates with the M₁TTM and M₂TTM radicals in addition to their intrinsic lower symmetry, which provides a boost to their PLQE (~2–10%). The fully substituted M₃TTM radical is of higher symmetry in its ground state than the less substituted M₁TTM and M₂TTM radicals and as such one would expect its PLQE to be low. However, in this case the excited state symmetry breaking is enhanced in M₃TTM as there is co-operative movement of one of the central rings (increased dihedral) and its corresponding mesityl group (decreased dihedral) in the relaxed D₁ state (Fig. 1b). This excited state symmetry breaking lowers the symmetry and increases the oscillator strength thus rationalizing the

enhanced emission of M₃TTM and its highest PLQE out of the MₓTTM series allowing it to circumvent its forbidden emission. Similar enhancement was observed experimentally for the D–A˙ structured radical M₂TTM-3PCz, whose emission can be described as a hybrid of short-range CT character localized within the TTM moiety and long-range CT character assigned to the D–A˙ interaction[32]. In these structures the short-range CT is enhanced by mesityl substitution (see the extended discussion in Supplementary Note 5).

## Device characterization

Given the promising photophysical properties of D–A˙ type radical M₂TTM-3PCz, which include high PLQE (Table 1) and excellent thermal- and photostability (Supplementary Figs. 8 and 44), its electroluminescence (EL) performance was evaluated. The optimized device configuration and energy level diagram of each layer are shown in Fig. 4a. The OLED has the following architecture: ITO/MoO₃ (3 nm)/TAPC (35 nm)/5 wt% M₂TTM-3PCz:CBP (30 nm)/B3PYMPM (35 nm)/LiF (0.8 nm)/Al (100 nm). M₂TTM-3PCz doped into CBP acts as the emitting layer and the doping concentration is 5 wt%. The maximum external quantum efficiency (EQE) for M₂TTM-3PCz OLEDs is 27.9% (Fig. 4b), significantly higher than previously recorded for OLEDs with TTM-3PCz as the emitter (EQE, 17%)[16]. The turn-on voltages are around 3 V (Fig. 4c), comparable to the previously reported TTM-3PCz OLEDs. The device displays deep-red emission with an EL peak at 689 nm, matching the PL spectrum of the light-emitting layer (Fig. 4d). The device shows maximum current efficiency of 59.1 cd A⁻¹ and maximum power efficiency of 60.9 lm W⁻¹. Thus, mesityl substituted D–A˙ type TTM radical derivatives can be used as excellent emitters for improved doublet OLEDs.

In conclusion, design of luminescent π-radicals is no longer restricted by practical synthetic aspects. We provide tools for quantitative synthesis and structural characterization of TTM radical and its D–A˙ backbone derivatives. The addition of mesityl substituents stabilizes the radical as well as its anionic form, which importantly prevents spontaneous oxidation and hydrogenation reactions, giving full synthetic control of the preparation of π-radicals. This also means that the radical conversion can be monitored using NMR spectroscopy. The mesityl groups serve another important function as sterically protecting the molecule, thereby enabling S–M couplings selectively at the non-mesitylated sites and clean synthesis of conjugated D–A˙ radicals. We find that mesitylation quenches the dominant nonradiative channels of TTM through excited-state symmetry breaking by as much as 6-fold from TTM to M₃TTM, meaning that highly symmetric TTMs can circumvent their intrinsic dark nature. Our studies on CBP films show that radical emitters can form emissive, long-lived intermolecular CBP

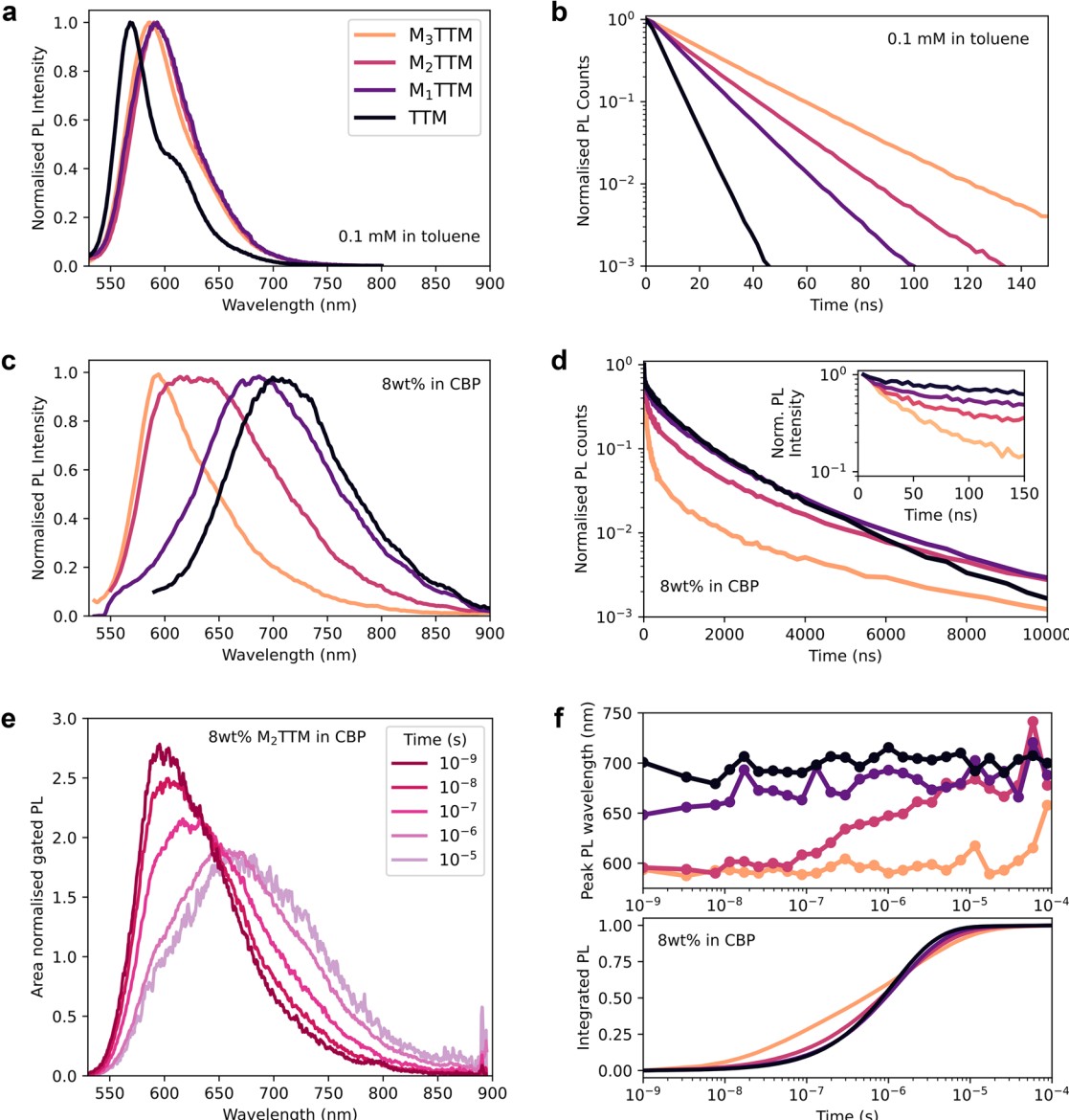

**Fig. 3 | Emission properties of the M$_x$TTM series. a** Steady-state emission in toluene solution and (**b**), emission kinetics in the 580–610 nm region showing rapid emission following 520 nm excitation. **c** Steady-state emission in 8 wt% evaporated films in CBP showing increasing emission red-shift and linewidth broadening with decreasing mesitylation and (**d**), total emission kinetics in the range of 550–880 nm

following 520 nm excitation using 100 fs pulses with fluence 5 μJ cm$^{-2}$. Inset shows early time kinetics. **e** Time-gated PL of 8 wt% M$_2$TTM in CBP film showing monomer emission at nanosecond times and exciplex emission (red-shifted by >0.2 eV) at microsecond times. **f** Dynamics of emission in CBP films showing time dependence of peak emission wavelength and integrated PL counts in the range of 550–880 nm.

(hole):radical (electron) charge-transfer states, the first experimental observation of the phenomenon for emissive radicals, and that the properties of this state can be controlled through increasing steric shielding of the radical. Mesityl substitution allows for the preparation of emissive main-chain polyradicals with solution and solid state PLQEs >10% at beyond 800 nm. Additionally, state-of-the-art OLED devices using mesityl substituted D−A$^\bullet$ type radical M$_2$TTM-3PCz as the emitter show a maximum EQE of 27.9%. Thus, our methodology unlocks the previously unexplored optoelectronic potential of TTM radical, providing unprecedented synthetic access to derivatives spanning small molecules to conjugated polymers with both record-breaking luminescence efficiencies and unexpected optical properties.

## Methods

### Characterization and techniques

NMR spectra were recorded on Bruker Avance 400 MHz ($^1$H, 400 MHz; $^{13}$C, 100 MHz) and 600 MHz ($^1$H, 600 MHz; $^{13}$C, 150 MHz) spectrometers.

Chemical shifts are reported in $\delta$ (ppm) relative to the solvent peak: chloroform-$d$ (CDCl$_3$: $^1$H, 7.26 ppm; $^{13}$C, 77.16 ppm), tetrahydrofuan-$d_8$ (THF-$d_8$: $^1$H, 3.58, 1.72 ppm; $^{13}$C, 67.21, 25.31 ppm) and dimethyl sulfoxide-$d_6$ (DMSO-$d_6$: $^1$H, 2.50 ppm; $^{13}$C, 39.52 ppm). For NMR monitored reactions the spectra were referenced against 1,3,5-trimethoxybenzene ($^1$H, 6.09 ppm) as the internal standard. Mass spectra were obtained using a Waters Xevo G2-S benchtop QTOF mass spectrometer (equipped with an atmospheric solids analysis probe, ASAP), a Thermo Finnigan Orbitrap Classic mass spectrometer (using direct injection) or a Bruker UltrafleXtreme MALDI TOF/TOF mass spectrometer at Yusuf Hamied Department of Chemistry, University of Cambridge. C, H, N combustion elemental analyses (EA) were obtained on an Exeter Analytical Inc. CE-440 elemental analyser and the results are reported as an average of two samples. Gel permeation chromatography (GPC) was carried out using an Agilent 1200 Series GPC-SEC System equipped with two sequential Phenogel™ 10 μm Linear(2) 300 × 7.8 mm LC columns. The eluent was chlorobenzene and the operating temperature was

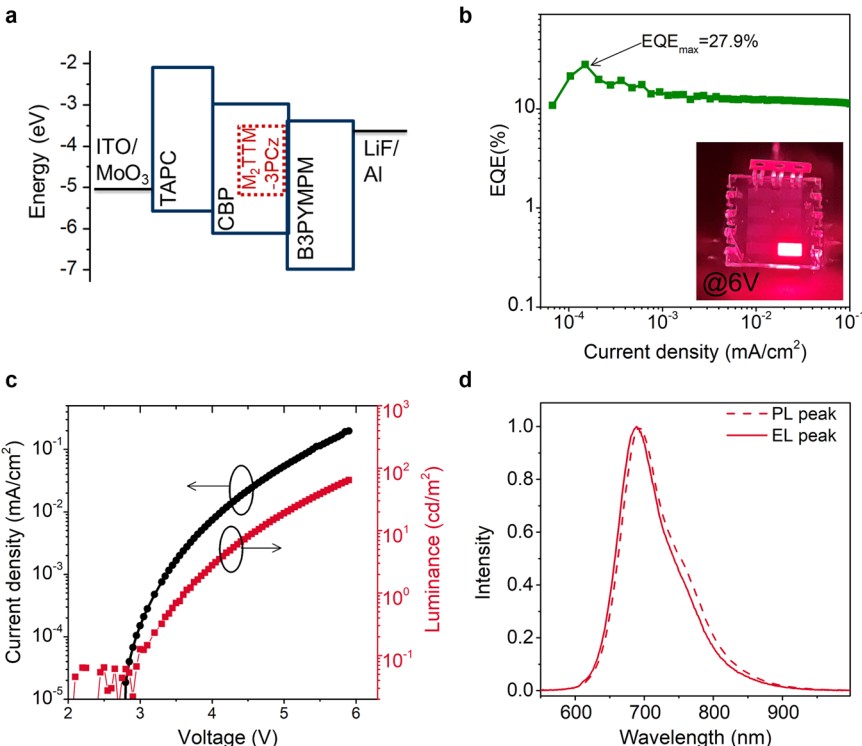

**Fig. 4 | Electrical and electroluminescent characteristics of OLEDs with D–A·**
**type radical M₂TTM-3PCz as the emitter. a** Structure of M₂TTM-3PCz based
OLEDs and energy level diagram of each layer. **b** EQE versus current density characteristic. Inset shows the optical image of radical OLEDs with M₂TTM-3PCz as
emitter. **c** Current density-voltage-luminance. **d** Normalized photoluminescence
spectrum of the light-emitting layer of 5 wt% M₂TTM-3PCz doped CBP film (photoexcited at 405 nm, dotted line) and electroluminescence spectrum of the device
at 0.2 mA cm⁻² (solid line).

80 °C. The number-average ($M_n$) and weight-average ($M_w$) molecular
weights were determined against polystyrene standards. Thermogravimetric analysis (TGA) was run under $N_2$ atmosphere at a heating
rate of 10 °C min⁻¹ using a Mettler Toledo TGA/DSC 2 instrument at a gas
flow of 125 cm³ min⁻¹. Flash chromatography was carried out using
Biotage® Isolera™ Four System and Biotage® SNAP/Sfär Silica flash
cartridges.

### X-Ray crystallography

Crystals were obtained by dissolving the sample in $CDCl_3$ in an NMR
tube. MeOH was added carefully on top and the solvents were allowed
to mix slowly in the dark. TTM, M₁TTM, M₂TTM and M₃TTM were collected as red single-crystals. Single-crystal X-ray diffraction (SXRD) data
were collected on a Bruker D8-QUEST diffractometer, equipped with an
Incoatec IμS Cu microsource ($\lambda$ = 1.5418 Å) and a PHOTON-III detector
operating in shutterless mode. The temperature was controlled by an
Oxford Cryosystems open-flow $N_2$ Cryostream operating at 180(2) K.
The control and processing software was Bruker *APEX4*. The diffraction
images were integrated using *SAINT* in *APEX4* and a multi-scan correction was applied using *SADABS*. The final unit-cell parameters were
refined against all reflections over the full data range. Structures were
solved using *SHELXT*[38] and refined using *SHELXL*[39]. All non-H atoms were
refined with anisotropic displacement parameters. H atoms were placed
in calculated positions and allowed to ride during subsequent refinement. Structure solution and refinement were largely straightforwards,
except as noted for M₃TTM in Supplementary Note 3.

### DFT calculations

The density functional theory (DFT) calculations were performed
using Gaussian 16 program. Ground state geometries were optimized
at UB3LYP/def2-SVP level (for radicals) or at B3LYP/def2-SVP level (for
αH precursors and CBP). The dispersion correction was conducted by

Grimme's D3 version[40]. Basing on the optimized ground state geometries, the vertical excitation energies were evaluated at UCAM-
B3LYP/def2-TZVP by time-dependent DFT (TD-DFT) treatment. The D₁
geometries were optimized at UCAM-B3LYP/def2-SVP level and the
relevant energies were evaluated by the same functional but basis set
of def2-TZVP. The D₁ state dihedrals were scanned by TD-DFT at
UCAM-B3LYP/def2-TZVP and UPBE0/Def2-TZVP levels. Excited states
analysis and the ADCH charge analysis were processed with the above
results using Multiwfn 3.8 program according to the program manual
and literature method[41]. The singly occupied molecular orbitals
(SOMOs) were assigned as occupied/unoccupied (SOMO/SUMO)
according to the distribution of the unpaired electron, which is to
describe the interaction between occupied–virtual molecular orbitals,
i.e., HOMO–SUMO and SOMO–LUMO, in the computational analysis.

The Einstein equation was used to estimate the radiative decay
rate ($k_r$)[31,42].

$$kr = \frac{\triangle E^3}{3\varepsilon_0 h^4 c^2} |\boldsymbol{\mu}_i|^2 \qquad (1)$$

where $\varepsilon_0$, $c$, $\Delta E$ and $\boldsymbol{\mu}_i$ denote vacuum permittivity, speed of light,
energy of D₁ state relative to D₀ state and transition dipole moment
(TDM) between the D₀ and D₁ states, respectively.

The value of TDM ($\boldsymbol{\mu}_i$) is related to oscillator strength ($f$) by the
relation[43,44]:

$$f = \frac{2m_e \triangle E}{3\hbar^2 e^2} |\boldsymbol{\mu}_i|^2 \qquad (2)$$

where $m_e$ is the mass of electron, $\Delta E$ is the energy of transition, $\hbar$ is the
reduced Planck's constant and $e$ is the charge of electron. Squared $\boldsymbol{\mu}_i$
can be obtained with $f$. Herein, the value of each component can be

determined by average direction of TDM (ratio) and squared $\boldsymbol{\mu}_i$ (sum of squares).

## Cyclic voltammetry

Cyclic voltammetry was carried out on PalmSens EmStat4S potentiostat in a three-electrode setup by using a glassy carbon (GC) or a platinum (Pt) electrode (3.0 mm diameter, for solution samples) or a Pt wire (for solid-state film samples) as the working electrode (WE), platinum wire as the counter electrode (CE) and freshly activated silver wire as the Ag/Ag$^+$ pseudoreference electrode (RE). The silver wire was activated by immersing in concentrated HCl solution to remove any silver oxides or other impurities, then rinsed with water and acetone and dried prior to each measurement. The RE was calibrated against ferrocene/ferrocenium (Fc/Fc$^+$) redox couple at the end of each measurement (the Fc/Fc$^+$ half-wave potential, $E_{1/2}$, was detected at 0.21 V vs. Ag/Ag$^+$ in THF electrolyte solution, at 0.45 V vs. Ag/Ag$^+$ in acetonitrile electrolyte solution and at 0.55 V vs. Ag/Ag$^+$ in DCM electrolyte solution). The supporting electrolyte was 0.1 M solution of Bu$_4$NPF$_6$ in anhydrous THF or in anhydrous DCM (for solution samples) or 0.1 M solution of Bu$_4$NPF$_6$ in anhydrous MeCN (for solid-state film samples). The electrolyte was bubbled with Ar gas before each measurement to romove any dissolved oxygen. For solution samples, the sample concentration was in was in the order of $10^{-5}$ M. For solid-state film samples, the Pt wire WE was coated with the polymer by drop-casting from 10 mg mL$^{-1}$ polymer in chlorobenzene solution.

## Steady state optical absorption and photoluminscence spectroscopy

Ultraviolet/visible/NIR spectra were measured with a commercially available Shimadzu UV-2550 spectrophotometer and a Shimadzu UV-1800 spectrophotometer. Photoluminescence was measured in a home-built setup by providing a continuous wave excitation at 532 nm using a diode laser. Photoluminscence is collected in a reflection mode setup after passing photons through a 550 long-pass filter (Thor Labs). The transmitted photons then are collected in a collimating 2-lens apparatus and directed into an optical fiber which supplies the photons into a calibrated grating-spectrometer (Andor SR-303i) and finally into a Si-camera where it is recorded. Output spectra are corrected taking into account the filter transmission and camera sensitivity. The excitation spectra were measured with a commercially available Edinburgh Instruments FS5 Spectrofluorometer system using a xenon lamp light source.

## Photoluminscence quantum efficiency

Steady-state PLQE measurements were performed using an integrating sphere. A continuous-wave 532 nm excitation is provided by a 532 nm diode laser with excitation powers of 10–300 mW cm$^{-2}$. A focussed beam of diameter 700 μm was used to excite the samples. The emission was directed using an optical fiber in a calibrated grating spectrometer (Andor SR-303i) onto a Si-camera.

## Time resolved single photon counting

The studied solution samples are irradiated using an electrically pulsed 532 nm laser using a function generator at a frequency of 20 MHz providing a time resolution of upto 100 ns. Photons emitted from the sample were passed through a 550 nm long-pass filter (Thor Labs Ltd.) to remove the laser scatter. The subsequently transmitted photons are then collected by a Si-based single-photon avalanche photodiode. The instrument response function was found to be ~0.1 ns in this setup.

## Transient photoluminscence spectroscopy

Transient photoluminescence spectra at nanosecond-microsecond timescales were recorded using an electronically gated intensified CCD (ICCD) camera (Andor iStar DH740 CCI-010) connected to a calibrated grating spectrometer (Andor SR303i). A narrowband non-colinear optical parametric amplifier pumped with a frequncy doubled output of a 1 kHz 800 nm laser pulse from a Ti:sapphire amplifier was used to generate a tunable 250-fs excitation pulse. Suitable long-pass filters (Edmund Optics) were used to prevent scattered laser signals from entering the spectrometer. Temporal evolution of the PL emission was obtained by stepping the ICCD delay with respect to the excitation pulse, with a minimum gate width of 5 ns. The raw data was corrected to account for filter transmission and camera sensitivity.

## M$_x$TTM:CBP films preparation and device fabrication

The M$_x$TTM doped CBP films and radical OLEDs were fabricated in a thermal evaporation system in a single high-vacuum chamber (Angstrom Engineering EvoVac 700 system) with a base pressure of less than $5 \times 10^{-7}$ Torr. Before deposition, the pure galss or ITO-coated glass substrates were sequentially cleaned ultrasonically with deionized water, acetone, and isopropanol, and followed by O$_2$ plasma for about 10 min. M$_x$TTM:CBP films were deposited on pure glass substrates by co-evaporation of M$_x$TTM and CBP materials. Analogous procedure was used for evaporated films of M$_x$TTM in diphenyl[4-(triphenylsilyl) phenyl]phosphine oxide (TSPO1).

The OLED architecture designed for M$_2$TTM-3PCz is indium tin oxide (ITO)/molybdenum trioxide (MoO$_3$) (3 nm)/TAPC (35 nm)/5 wt% M$_2$TTM-3PCz in CBP (30 nm)/B3PYMPM (35 nm)/lithium fluoride (LiF) (0.8 nm)/aluminum (100 nm). MoO$_3$ was deposited at 0.1 Å s$^{-1}$ and organic layers were deposited at a rate of 0.5–1 Å s$^{-1}$. LiF was subsequently deposited at 0.1 Å s$^{-1}$ and then capped with Al (~1 Å s$^{-1}$). The devices were exposed once to nitrogen gas after the formation of the organic layers because a metal mask was included to define the cathode area. 1,1-Bis[(di-4-tolylamino)phenyl]cyclohexane (TAPC)[45] and 4,6-bis(3,5-di(pyridin-3-yl)phenyl)-2-methylpyrimidine (B3PYMPM)[46] are used as the hole- and electron-transport layers, respectively. We note that TAPC and B3PYMPM can effectively confine holes and electrons within the exciton recombination zone. On the one hand, TAPC prevent electrons transporting across the emissive layer because of its shallower LUMO energy level than that of CBP or the SOMO energy level of M$_2$TTM-3PCz, thus substantially minimizing electron leakage from the emissive layer. On the other hand, a large energy offset of the HOMO energy level between the CBP and B3PYMPM prevent the injected holes transferring into the nonradiative layer.

For all the OLEDs, the 4.5 mm$^2$ effective active area is defined by the geometrical overlap of the two electrodes. All the current density, voltage and electroluminescence characteristics were performed at room temperature in ambient air, using a Keithley 2400 source meter, a Keithley 2000 multimeter and a calibrated silicon photodiode. The EL and PL spectra were measured by the same setup used in photoluminscence spectroscopy.

## Data availability

The computational atomic coordinates of the optimized structures and the optical spectroscopy and device characterization data generated in this study have been deposited in the University of Cambridge Repository under accession code https://doi.org/10.17863/CAM.97032. The X-ray crystallographic coordinates for structures reported in this study have been deposited at the Cambridge Crystallographic Data Centre (CCDC) under deposition numbers 2195365–2195368. These data can be obtained free of charge from CCDC via www.ccdc.cam.ac.uk/data_request/cif. The authors declare that all other data that support the findings of this study are available within the article and its supplementary information files. Source data are provided with this paper.

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

## Acknowledgements

We thank Dr Andrew Bond for carrying out the X-ray crystallography measurements and data analysis at the Yusuf Hamied Department of Chemistry, University of Cambridge. P.M., R.C. and W.Z. have received funding from the European Union's Horizon 2020 research and innovation programme under the Marie Skłodowska-Curie grant agreements No. 891167, No. 859752 and No. 886066. We acknowledge funding from the European Research Council under the European Union's Horizon 2020 research and innovation programme grant agreement No. 101020167 (R.H.F., P.M., E.G., S.G.) and the Engineering and Physical Sciences Research Council NanoDTC, EP/S003126/1, EP/S022953/1 (S.G.).

## Author contributions

P.M. designed, synthesized and characterized the materials. R.C., S.G. and B.L. conducted the optical spectroscopy measurements and data analysis. E.G. and Y.S. fabricated and characterized the OLED devices. W.Z. carried out the quantum-chemical calculations and data analysis. H.F. performed the thermal analysis. R.H.F. and H.B. supervised the work. All authors discussed the results and contributed to writing the manuscript.

## Competing interests

The authors declare no competing interests.
