## [Peer Review File · Nature Communications]

Mesitylated trityl radicals, a platform for doublet emission: symmetry breaking, charge-transfer states and conjugated polymersREVIEWER COMMENTS

Reviewer #1 (Remarks to the Author):

Murto et al reported a methodology for synthesis of neutral radical emitters applicable in OLED devices. The radicals enable emission from doublet states, bypassing the issue related to harvesting of triplet excitons in current light-emitting materials. In my opinion, this paper is interesting, and provides important insights into the nature / mechanism of materials properties / performance from an experimental point of view. This paper can provide guidelines to tune radical emitters for advancing radical-based OLEDs.

However, this manuscript requires a major revision before I can recommend its publication in Nature Communications.

Major issues:

(1) On page 9, the authors indicated the possibility of spontaneous excited-state symmetry breaking in M3TTM, which can increase the PL enhancement. A computational investigation by Bredas et al shows that there is a cosine-type correlation between the radiative decay rates (k_r) and the dihedral angle (Φ) between the donor and acceptor moieties (<https://doi.org/10.1002/adfm.202002916>). The same aspect of structural (D-A dihedral) fluctuations may be valid for the new emitters studied in the paper. The above paper needs to be cited, and a discussion on this aspect should be provided.

(2) On page 11, the authors pointed to a remarkable red emission and microsecond lifetimes for the emitters when doped in CBP hosts. In a previous computational study, it is shown that CPB indeed remarkably modulates the electronic properties of radical emitters (<https://doi.org/10.1039/D2CP00592A>). The same study shows that the D1 excitons of emitter-CBP clusters present a significant "intermolecular" CT character with red shift. The radiative decay rates of the emitter-CBP clusters is an order of magnitude lower than that for isolated emitters, which is in agreement with the report of the current manuscript submitted to Nature Communications. Although the paper is already given as Ref 32, it needs to be cited on the right page with additional discussion to acknowledge the previous paper's reports.

(3) Authors are talking about hole transfer to CBP. Is this not more about electron transfer from CBP to the radical emitters, and possibly forming exciplexes?

(4) I suggest authors provide some info on the radiative decay rates and EQE.

Minor issues:

Page 1, in the abstract, MxTTM are given. "M" needs to be defined when it appears for the first time in the manuscript.

Page 5, "S-M coupling" might be not obvious for general readers, and needs to be defined.

Please check numbers of the supplementary figures to match with the main text.

Page 15, either remove "the first time" or indicate this is the first experimental observation in the following sentence "...long-lived intermolecular CBP (hole):radical (electron) charge-transfer states, the first time such a phenomenon has been reported for emissive radicals.."

Reviewer #2 (Remarks to the Author):

This manuscript by Murto et al. titled "Mesitylated trityl radicals is a new platform for doublet emission" reports a new synthetic methodology for preparation of organic luminescent n-radicals materials, specifically mesityl substituted tris(2,4,6-trichlorophenyl)methyl (TTM) radical. The authors have conducted a systematic study focusing on the radical characterization, photophysical property, theoretical computation, electrochemistry (cyclic voltammetry) and device performance. Through the facile syntheses, such as mesitylation and donor-acceptor modification, the authors claimed to successfully stabilize the radical as well as boost the PLQYs of the titled compounds, with the device's EQE reached 27 %. This work is interesting. However, the previous relevant publications make the reviewer skeptical about its novelty and urgency, which may not meet the gold standard for its publication on the journal Nature Communication.

Firstly, in terms of structural novelty and device performance, a similar TTM analog (TTM-3NCz) with a comparatively high EQE (27 %) has already been published by the same authors (Nature 563, 536-540 (2018)). Therefore, the high EQE claimed in the current study has already been achieved by TTM-3NCz that is NOT protected by mesityl groups.

Secondly, in terms of fundamental photophysics, the of importance on the intermolecular charge transfer phenomenon (exciplex) was not well explained and/or assigned by sufficient steady-state and time-resolved measurements. What does the exciplex look like? What is the driving force (motion) and energy barrier for the exciplex formation? Besides steric effect, does the band alignment also affect the kinetics of intermolecular charge transfer? None of these crucial issues were resolved by the authors. Furthermore, based on the onset of absorption and CV spectra, the band alignments between CBP and MxTTM are in type I configuration. How to generate any red-shifted emission comparing to MxTTM alone was not explained. The lack of fundamental assessment and the self-contradictory assignments weaken the readability of this paper.

Comments and suggestion:

1. The PL spectra of MxTTM in CBP matrix were red-shifted comparing to that in toluene, which was assigned by the authors as the emission from intermolecular charge transfer species (exciplex). However, considering the band alignment between CBP and MxTTM, the LUMO of CBP is higher in energy than that of MxTTM, while the HOMO of the former is lower than that of the latter. In this case, upon photoexcitation on CBP, the electron populated at LUMO on CBP would transfer to the LUMO of MxTTM, followed by transition (fluorescence) of MxTTM itself. Therefore, one should not perceive any red-shifted emission for type 1 exciplexes. Photophysical measurements on a neat film (MxTTM) would help. Besides, the authors could choose other common host materials (with different band alignment) to confirm the exciplex emission.

2. In order to confirm the photophysical purity of all studied compounds, the authors are required to provide excitation spectra monitored at different wavelength of the emission spectra. Please check whether the excitation and absorption spectra resemble in shape.

3. The authors have performed photostability tests on the OLEDs in previous reports (Nature 563, 536- 540 (2018)). As a comparison, it is recommended to compare the device photostability with and without the mesityl group.

4. What are the monitored wavelength of the kinetic traces in Fig. 3b and 3d?

5. The authors have observed a temporal evolution depicted in Fig. 3e. As a result, there might exist a pseudo precursor-successor relationship in the MxTTM@CBP systems. Additionally, the authors also stated that the spectral shift is dependent on temperature. If the red-shifted emission does not belong

to exciplex formation (as mentioned in query #1), what kind of slow excited-state motion (nanosecond scale) contributes to the spectral shift? What is the temperature dependence or energy barrier (Arrhenius plot) of this motion?

6. In page 12, line 231, the authors claimed that the intermolecular charge transfer could be sterically controlled. However, the exciplex formation could also be thermodynamically controlled. The authors are encouraged to add the band alignment diagram in the supporting information.

Reviewer #3 (Remarks to the Author):

The authors show new design of stable luminescent radicals based on an idea to cap the p-positions of TTM radical using mesityl groups. Radical conversion and alpha-hydrogenation processes were observed by ¹H NMR. Basic photophysical properties of M1TTM, M2TTM, and M3TTM were measured in toluene solutions, and the increased photoluminescence quantum efficiency of M3TTM was explained by symmetry breaking in the excited state. Luminescent behaviours of these radicals in CBP films were observed by nanosecond spectroscopy. The authors also report clean synthesis of D-A type radical using M2TTM precursor, which show higher photoluminescence quantum yields than the one made using TTM precursor. Practically, highly efficient OLED was performed using this radical. Diboronic acid pinacol ester was prepared from M1TTM precursor, and NIR luminescent main-chain conjugate polyradicals were synthesized.

This work is an assembly of various noteworthy results in the field of luminescent materials, and they are supported by plenty of measurements and computational data. This smart synthetic methodology of these radicals will have many applications in the future.

Comments:

1. The authors claim that red-shifted emissions of TTM, M1TTM, and M2TTM in CBP film is due to intermolecular charge transfer from CBP donor, however the possibility of intermolecular excimer formation between the radicals are not excluded. The concentration of 8wt% is relatively high (please see the series of works by S. Kimura et al. for example *Angew. Chem. Int. Ed.* 2018, 57, 12711-12715). If you want to prove the charge transfer from CBP, you can show the concentration dependence of the luminescence.

2. Highly photoluminescent symmetric TTM derivatives have already been reported (S. Castellanos et al. *J. Org. Chem.* 2008, 73, 3759-3767). This paper should be mentioned in the manuscript.

3. The idea to substitute p-positions of triarylmethyl radical have recently reported using PyBTM (Y. Hattori et al. *Chem. Sci.* 2022, 13, 13418-13425), although this seems more like D-A type radical. It might be better to cite this paper for comparison as well to make Fig. S1 more meaningful. Electrochemical potentials of PyBTM, bisPyTM, and trisPyM were reported in the literatures.

4. In the Table 1, it seems that some absorption and emission wavelengths are intentionally expressed in multiples of fives, and the others are not. Consistency or captions may be needed.

5. Introduction (line 34), "almost exclusively" sounds a bit too strong expression considering known several works of other luminescent radicals.

6. In Page 9 (line 178), it is more reader-friendly to write photoluminescence quantum yield of TTM-3PCz for comparison.

7. Fig. S5, PFTTM and PCzTTM would be typos of PFMTTM and PCzMTTM.

Point-by-point response

Reviewer #1 (Remarks to the Author):

Murto et al reported a methodology for synthesis of neutral radical emitters applicable in OLED devices. The radicals enable emission from doublet states, bypassing the issue related to harvesting of triplet excitons in current light-emitting materials. In my opinion, this paper is interesting, and provides important insights into the nature / mechanism of materials properties / performance from an experimental point of view. This paper can provide guidelines to tune radical emitters for advancing radical-based OLEDs.

However, this manuscript requires a major revision before I can recommend its publication in Nature Communications.

Major issues:

Comment 1

On page 9, the authors indicated the possibility of spontaneous excited-state symmetry breaking in M3TTM, which can increase the PL enhancement. A computational investigation by Bredas et al shows that there is a cosine-type correlation between the radiative decay rates (k_r) and the dihedral angle (Φ) between the donor and acceptor moieties (<https://doi.org/10.1002/adfm.202002916>). The same aspect of structural (D-A dihedral) fluctuations may be valid for the new emitters studied in the paper. The above paper needs to be cited, and a discussion on this aspect should be provided.

Response 1

We are grateful to the reviewer for the good point. Detailed relationship between the radiative decay rates and the dihedral angle between the mesityl groups and the TTM core has been added in Supplementary Fig. 23. Relevant discussion has been added in the main manuscript and we have cited the suggested paper as ref. 31 in the optical spectroscopy section as follows:

“As developed below, this may arise from spontaneous excited-state symmetry breaking that is distinctively different from typical CT emission arising from D–A* hybrids in correlation between radiative decay rates and dihedral angles (see Supplementary Fig. 23 and the extended discussion in Supplementary Section 7).³¹”

Following the reviewer’s point, we have added further discussion with respect to the previous study of Bredas, et al. (<https://doi.org/10.1002/adfm.202002916>). To keep the main manuscript concise, we have added the extended discussion in Supplementary Section 7 as follows:

“**Radiative decay rate of M₃TTM.** To further analyse the emission nature of M₃TTM, we scanned models based on its optimized D₁ state geometry with manually set dihedral angle between one of the dichlorobenzene ligands and its corresponding mesityl group from 0 to 90° while the other two dihedrals were fixed at 90°. The radiative decay rate was estimated by the Einstein equation^{12,13} with calculated excited state energy levels and relevant oscillator strengths (Supplementary Section 2).^{14,15} As shown in Fig. S23, the radiative decay rates were about an order of magnitude lower than those of CT type radical emitters,^{13,24} in line with the experimental decay rates (Table 1 in the main manuscript), due to the excited state transition dipole moment cancellation of the highly symmetric TTM core. The radiative decay rates were higher when the geometry was less symmetric, which is observed as relaxation from a darker D₀ geometry to a brighter D₁ geometry.”

Comment 2

On page 11, the authors pointed to a remarkable red emission and microsecond lifetimes for the emitters when doped in CBP hosts. In a previous computational study, it is shown that CPB indeed remarkably modulates the electronic properties of radical emitters (<https://doi.org/10.1039/D2CP00592A>). The same study shows that the D1 excitons of emitter–CBP clusters present a significant “intermolecular” CT character with red shift. The radiative decay rates of the emitter–CBP clusters is an order of magnitude lower than that for isolated emitters, which is in agreement with the report of the current manuscript submitted to Nature Communications. Although the paper is already given as Ref 32, it needs to be cited on the right page with additional discussion to acknowledge the previous paper’s reports.

Response 2

We agree with the reviewer’s point on the excellent agreement of our experimental results and previous computational studies (<https://doi.org/10.1039/D2CP00592A>). We have highlighted the relevant works as refs. 31 and 32 more appropriately in the optical spectroscopy section:

“We consider that the red-shifted and long-lived emission is from intermolecular charge transfer states with CBP acting as electron donor, in agreement with previous computational studies of Abroshan, et al.^{31,32} on radical emitter:CBP host clusters.”

Comment 3

Authors are talking about hole transfer to CBP. Is this not more about electron transfer from CBP to the radical emitters, and possibly forming exciplexes?

Response 3

We have described hole transfer to CBP because, following direct excitation of M_x TTM radical at 520 nm, the radical can either give monomolecular emission from the D_1 state or produce an intermolecular charge transfer state. We have illustrated the latter scenario in Supplementary Fig. 2. It is reasonable to assume that electron is localised on the electron-deficient radical, while hole is the mobile component of the charge transfer state in the majority CBP host, based on both the band alignment (Supplementary Fig. 2) and previous studies on radical emitter:CBP blends (Nature 563, 536–540 (2018)). Thus we have highlighted the process from the point of view of the hole.

Comment 4

I suggest authors provide some info on the radiative decay rates and EQE.

Response 4

Following the reviewer's suggestion, we have added the radiative and nonradiative decay rates in Table 1. We note that mesityl substitution systematically increases the radiative decay rate and suppresses the nonradiative decay rate. Mesitylation also enhances the radiative decay rate of M_2 TTM-3PCz as highlighted in the optical spectroscopy section:

“We note that mesityl substitution also enhances the PLQE of D–A* type radical M_2 TTM-3PCz from 46% (for TTM-3PCz) to 84% in toluene solution, which stems from the increase in radiative decay rate (Table 1 and Supplementary Fig. 34).¹⁶”

We have also provided the EQE in the device characterization section:

“The maximum external quantum efficiency (EQE) for M_2 TTM-3PCz OLEDs is 27.9% (Fig. 4b), significantly higher than previously recorded for OLEDs with TTM-3PCz as the emitter (EQE, 17%).¹⁶”

Minor issues:

Comment 5

Page 1, in the abstract, MxTTM are given. “M” needs to be defined when it appears for the first time in the manuscript.

Response 5

Following the reviewer’s comment, we have defined “mesityl (M)” when it first appears in the abstract and introduction section.

Comment 6:

Page 5, “S–M coupling” might be not obvious for general readers, and needs to be defined.

Response 6

Following the reviewer’s comment, we have defined “Suzuki–Miyaura (S–M) cross-coupling” when it first appears in the introduction section.

Comment 7

Please check numbers of the supplementary figures to match with the main text.

Response 7

We have checked through the figure numbering in the main manuscript and supplementary information files.

Comment 8

Page 15, either remove “the first time” or indicate this is the first experimental observation in the following sentence “...long-lived intermolecular CBP (hole):radical (electron) charge-transfer states, the first time such a phenomenon has been reported for emissive radicals..”

Response 8

Following the reviewer’s suggestion, we have revised the sentence as follows:

“...long-lived intermolecular CBP (hole):radical (electron) charge-transfer states, the first experimental observation of the phenomenon for emissive radicals ...”

Reviewer #2 (Remarks to the Author):

This manuscript by Murto et al. titled “Mesitylated trityl radicals is a new platform for doublet emission” reports a new synthetic methodology for preparation of organic luminescent π -radicals materials, specifically mesityl substituted tris(2,4,6-trichlorophenyl)methyl (TTM) radical. The authors have conducted a systematic study focusing on the radical characterization, photophysical property, theoretical computation, electrochemistry (cyclic voltammetry) and device performance. Through the facile syntheses, such as mesitylation and donor-acceptor modification, the authors claimed to successfully stabilize the radical as well as boost the PLQYs of the titled compounds, with the device’s EQE reached 27 %. This work is interesting. However, the previous relevant publications make the reviewer skeptical about its novelty and urgency, which may not meet the gold standard for its publication on the journal Nature Communication.

Firstly, in terms of structural novelty and device performance, a similar TTM analog (TTM-3NCz) with a comparatively high EQE (27 %) has already been published by the same authors (Nature 563, 536-540 (2018)). Therefore, the high EQE claimed in the current study has already been achieved by TTM-3NCz that is NOT protected by mesityl groups.

Secondly, in terms of fundamental photophysics, the importance on the intermolecular charge transfer phenomenon (exciplex) was not well explained and/or assigned by sufficient steady-state and time-resolved measurements. What does the exciplex look like? What is the driving force (motion) and energy barrier for the exciplex formation? Besides steric effect, does the band alignment also affect the kinetics of intermolecular charge transfer? None of these crucial issues were resolved by the authors. Furthermore, based on the onset of absorption and CV

spectra, the band alignments between CBP and MxTTM are in type I configuration. How to generate any red-shifted emission comparing to MxTTM alone was not explained. The lack of fundamental assessment and the self-contradictory assignments weaken the readability of this paper.

Response to the general comments

We thank the reviewer for providing valuable feedback. In response to the first point, we agree that the device EQEs of M₂TTM-3PCz and structurally related TTM-3NCz are comparatively high (the latter containing 9-(naphthalen-2-yl)-9*H*-carbazole donor). However, in this study we have specifically built upon TTM-3PCz radical with a 9-phenyl-9*H*-carbazole donor as it has shown significantly lower EQE of 17% (Nature 563, 536–540 (2018)). Mesitylated M₂TTM-3PCz almost doubles the EQE to 28%, which stems from enhancement of photoluminescence quantum efficiency from 46% (for TTM-3PCz; Nature 563, 536–540 (2018)) to 84% in toluene solution. We report mesityl substitution as steric protection that enables defect-free synthesis of radical derivatives spanning small molecules and D–A* systems to conjugated polyradicals. We also find that mesityl substitution systematically enhances the emission of different radical families, as exemplified by the state-of-the-art M₂TTM-3PCz radical OLED.

In response to the second point, we have thoroughly revised the intermolecular charge transfer concept and provided additional experimental data throughout the manuscript, as detailed in our responses to the reviewer's comments below. We have added a new Supplementary Fig. 2 to illustrate the band alignment, where driving force for charge transfer is the offset between CBP HOMO and radical HOMO energy levels with respect to radical SOMO energy level (type 2 band alignment between CBP HOMO/LUMO and radical HOMO/SOMO). Further photophysical measurements were carried out to estimate the energy barrier (Supplementary

Fig. 38). Charge transfer exciplex is not observed in TSPO1 host as its HOMO energy level is lower than that of the radical (type 1 band alignment). We have added this new data in Supplementary Fig. 7. By varying the radical concentration and comparing two different host:radical band alignments, we have confirmed the formation of intermolecular charge transfer states specifically in CBP and excluded the possibility of red-shifted emission coming from intermolecular excimers.

Comments and suggestion:

Comment 1

The PL spectra of MxTTM in CBP matrix were red-shifted comparing to that in toluene, which was assigned by the authors as the emission from intermolecular charge transfer species (exciplex). However, considering the band alignment between CBP and MxTTM, the LUMO of CBP is higher in energy than that of MxTTM, while the HOMO of the former is lower than that of the latter. In this case, upon photoexcitation on CBP, the electron populated at LUMO on CBP would transfer to the LUMO of MxTTM, followed by transition (fluorescence) of MxTTM itself. Therefore, one should not perceive any red-shifted emission for type 1 exciplexes. Photophysical measurements on a neat film (MxTTM) would help. Besides, the authors could choose other common host materials (with different band alignment) to confirm the exciplex emission.

Response 1

We thank the reviewer for pointing out this important aspect. We agree that the LUMO of CBP is higher in energy than that of MxTTM radicals. However, it has been demonstrated previously that photoexcitation of donor–acceptor type radicals such as TTM-1Cz (A. Abdurahman, et al.,

Nat. Mater. 19, 1224–1229 (2020); DOI 10.1038/s41563-020-0705-9) results in charge transfer from the HOMO of the donor to the SOMO of the radical because the HOMO of the radical is lower in energy than that of the donor (type 2 band alignment). The HOMO of CBP is similarly high in energy with respect to the HOMO of M_x TTM radicals. We have illustrated this type 2 alignment in Supplementary Fig. 2. As for the red-shifted emission in CBP: M_x TTM blends, the radical is photoexcited directly at 520 nm. The radical can give monomolecular emission from the D_1 state or, following hole transfer to the majority CBP host, produce a charge transfer state with electron on the radical. We describe this state as an intermolecular analogue of the intramolecular CT state in donor–acceptor radicals. We attribute the long-lived and red-shifted emission to these charge-transfer excitons, as illustrated in Supplementary Fig. 2.

Following the reviewer’s suggestion, we have made blend films of M_x TTM radical in TSPO1 whose HOMO energy level is lower than that of M_x TTM, hence type 1 band alignment (S. O. Jeon, et al., Adv. Mater. 23, 1436–1441 (2011); DOI 10.1002/adma.201004372). Comparison of blends of M_1 TTM and M_2 TTM in either TSPO1 or CBP (Supplementary Fig. 7) shows that TSPO1 does not facilitate intermolecular charge transfer and the emission is not red-shifted compared to the radical emission in toluene solution. On the other hand, Supplementary Fig. 7 also demonstrates that the emission red-shift in CBP is independent of blend concentration. We expect significant luminescence quenching in neat films, based on prior work on TTM radicals (Nature 563, 536–540 (2018); Nat. Mater. 19, 1224–1229 (2020)). We have referred to the new data in the optical spectroscopy section in the main manuscript as follows:

“The red-shift is independent of doping concentration and specifically observed in CBP films (Supplementary Fig. 7), excluding the possibility of formation of long-lived intermolecular excimers (Supplementary Fig. 36 and 37).^{33,34}”

Comment 2

In order to confirm the photophysical purity of all studied compounds, the authors are required to provide excitation spectra monitored at different wavelength of the emission spectra. Please check whether the excitation and absorption spectra resemble in shape.

Response 2

Following the reviewer's suggestion, we have measured the excitation spectra of all materials and compared them to the absorption spectra in Supplementary Fig. 33. For M_x TTM series, for example, we find that the two spectra resemble in shape and similar changes in the spectral profile can be found in both absorption and excitation spectra when increasing the mesitylation from TTM to M_3 TTM. We have referred to this data in the optical spectroscopy section as follows:

“Excitation spectra of the series follow the same trend observed in their absorption spectra (Supplementary Fig. 33).”

Comment 3

The authors have performed photostability tests on the OLEDs in previous reports (Nature 563, 536-540 (2018)). As a comparison, it is recommended to compare the device photostability with and without the mesityl group.

Response 3

Following the reviewer's suggestion, we have measured the photostability of the device active layer and reported the results in Supplementary Fig. 8. Both the spectral shape and the peak intensity remain virtually unchanged during 100 000 s (ca. 28 h) continuous wave excitation at 532 nm under ambient conditions, an order of magnitude longer than measured previously for

non-mesitylated radical (Nature 563, 536–540 (2018)). We have referred to the photostability data in the device characterization section:

“Given the promising photophysical properties of D–A* type radical M₂TTM-3PCz, which include high PLQE (Table 1) and excellent photostability (Supplementary Fig. 8)...”

Comment 4

What are the monitored wavelength of the kinetic traces in Fig. 3b and 3d?

Response 4

We thank the reviewer for pointing this out. The kinetics were monitored in the range of 550–880 nm for CBP film samples (Fig. 3d) and similarly for the integrated PL (Fig. 3f), whereas for the toluene solution samples the kinetics were monitored in the 580–610 nm region (Fig. 3b). We have added the wavelength ranges in the figure caption.

Comment 5

The authors have observed a temporal evolution depicted in Fig. 3e. As a result, there might exist a pseudo precursor-successor relationship in the M_xTTM@CBP systems. Additionally, the authors also stated that the spectral shift is dependent on temperature. If the red-shifted emission does not belong to exciplex formation (as mentioned in query #1), what kind of slow excited-state motion (nanosecond scale) contributes to the spectral shift? What is the temperature dependence or energy barrier (Arrhenius plot) of this motion?

Response 5

In response to the reviewer’s question about exciplex formation in CBP:M_xTTM systems, we refer to our response to query #1 where we have described the formation of the intermolecular

charge transfer state and provided additional data on CBP concentration series and TSPO1 host system. It is reasonable to assume that in the charge transfer state electron is localised on the electron-deficient radical while hole is the mobile component in the majority CBP host, based on both the band alignment (Supplementary Fig. 2) and previous studies on radical emitter:CBP blend systems (Nature 563, 536–540 (2018)). This charge-transfer exciton can either give red-shifted emission or regenerate the molecular exciton. Luminescence spectroscopy reveals that when the temperature is lowered, the charge transfer state responsible for the observed red-shifted emission is no longer able to reform the intramolecular state readily. We can estimate the activation barrier for this process in M₃TTM based on the emission rate of monomolecular emission sampled within the 570–590 nm spectral region. An Arrhenius analysis gives a value of 59 ± 13 meV. We have added a new Supplementary Fig. 38 to describe this process and commented it in the optical spectroscopy section:

“We propose that there is competition between direct intermolecular CT emission and endothermic regeneration of the molecular exciton, with an energy barrier of 59 ± 13 meV estimated for M₃TTM (Supplementary Fig. 38).”

Comment 6

In page 12, line 231, the authors claimed that the intermolecular charge transfer could be sterically controlled. However, the exciplex formation could also be thermodynamically controlled. The authors are encouraged to add the band alignment diagram in the supporting information.

Response 6

Following the reviewer’s valuable comment, we have added a band alignment diagram of the radical and CBP in Supplementary Fig. 2. For simplicity, we have only shown the calculated

and experimental band alignment of M₂TTM in Supplementary Fig. 2, as the alignment remain analogous for the entire M_xTTM series. Details of the energy levels of the M_xTTM series can be found in Supplementary Fig. 1. Discussion has been extended in the main manuscript and more data has been added in supplementary sections 7 and 8. We agree with the reviewer that the charge transfer emission is also thermodynamically controlled, as observed in temperature dependent transient luminescence spectroscopy (Supplementary Fig. 40–43). We have added a comment on this in the main manuscript in the optical spectroscopy section:

“This is the first observation of intermolecular charge-transfer state emission in TTM radicals, and we show its kinetics can be controlled both thermodynamically and sterically by mesitylation of the TTM moiety.”

Reviewer #3 (Remarks to the Author):

The authors show new design of stable luminescent radicals based on an idea to cap the p-positions of TTM radical using mesityl groups. Radical conversion and alpha-hydrogenation processes were observed by ¹H NMR. Basic photophysical properties of M1TTM, M2TTM, and M3TTM were measured in toluene solutions, and the increased photoluminescence quantum efficiency of M3TTM was explained by symmetry breaking in the excited state. Luminescent behaviours of these radicals in CBP films were observed by nanosecond spectroscopy. The authors also report clean synthesis of D-A type radical using M2TTM precursor, which show higher photoluminescence quantum yields than the one made using TTM precursor. Practically, highly efficient OLED was performed using this radical. Diboronic acid pinacol ester was prepared from M1TTM precursor, and NIR luminescent main-chain conjugate polyradicals were synthesized.

This work is an assembly of various noteworthy results in the field of luminescent materials, and they are supported by plenty of measurements and computational data. This smart synthetic methodology of these radicals will have many applications in the future.

Comments:

Comment 1

The authors claim that red-shifted emissions of TTM, M1TTM, and M2TTM in CBP film is due to intermolecular charge transfer from CBP donor, however the possibility of intermolecular excimer formation between the radicals are not excluded. The concentration of 8wt% is relatively high (please see the series of works by S. Kimura et al. for example *Angew.*

Chem. Int. Ed. 2018, 57, 12711-12715). If you want to prove the charge transfer from CBP, you can show the concentration dependence of the luminescence.

Response 1

Following the reviewer's valuable comment, we have measured the emission of TTM, M₁TTM and M₂TTM at lower concentrations down to 1 wt% in CBP. We have added the new spectra in Supplementary Fig. 7. The lower concentration spectra (Supplementary Fig. 7) are analogous to the higher concentration spectra (Fig. 3c, main manuscript) with no change in emission peak wavelengths, thus confirming that the emission is not dependent on the concentration. The sterically protected M₃TTM radical does not show redshifted emission at room temperature even at high concentration (Fig. 3c, main manuscript). As further evidence of intermolecular charge transfer state, we have compared the blends of radical in a wide band gap host TSPO1 (Supplementary Fig. 7) which does not facilitate intermolecular charge transfer and the emission is not red-shifted. This comparison also confirms that the red-shifted emission is not from radical-radical excimer as in the work of S. Kimura, et al. (Angew. Chem. Int. Ed. 2018, 57, 12711–12715). More discussion has been added in the optical spectroscopy section and the works of S. Kimura, et al. have been cited as refs. 33 and 34.

“The red-shift is independent of doping concentration and specifically observed in CBP films (Supplementary Fig. 7), excluding the possibility of formation of long-lived intermolecular excimers (Supplementary Fig. 36 and 37).^{33,34}”

Comment 2

Highly photoluminescent symmetric TTM derivatives have already been reported (S. Castellanos et al. J. Org. Chem. 2008, 73, 3759-3767). This paper should be mentioned in the

manuscript.

Response 2

We have added description on the mechanism of charge-transfer systems in the introduction section and mentioned the important work of S. Castellanos, et al. (J. Org. Chem. 2008, 73, 3759-3767; ref. 17) in the same context as follows:

“The degeneracy of the lowest energy excitations is alleviated with a dominant HOMO (donor) to SOMO (radical) electronic transition, thus the HOMO energy level of the donor and the SOMO energy level of the radical define the optical energy gap (Supplementary Fig. 2). Non-alternant donors with a strong CT character have been utilised to make even symmetric D–A* radicals highly emissive.¹⁷”

Comment 3

The idea to substitute p-positions of triarylmethyl radical have recently reported using PyBTM (Y. Hattori et al. Chem. Sci. 2022, 13, 13418-13425), although this seems more like D-A type radical. It might be better to cite this paper for comparison as well to make Fig. S1 more meaningful. Electrochemical potentials of PyBTM, bisPyTM, and trisPyM were reported in the literatures.

Response 3

Following the reviewer’s valuable comment, we have mentioned the work of Y. Hattori, et al. (Chem. Sci. 2022, 13, 13418-13425; ref. 22) in the introduction section as follows:

“Importantly, substitution of TTM structure with bulky mesityl (M) groups, first introduced into trityl radicals by Hattori, et al.,²² enables clean and selective synthesis of three families of radicals...”

We have also updated Fig. S1 and added the missing electrochemical potentials from literature. We have cited the published works as Supplementary refs. 1–4 in the revised supplementary information file.

Comment 4

In the Table 1, it seems that some absorption and emission wavelengths are intentionally expressed in multiples of fives, and the others are not. Consistency or captions may be needed.

Response 4

Following the reviewer's comment, we have provided detailed readings of the absorption and emission peak wavelengths in Table 1 in the revised manuscript.

Comment 5

Introduction (line 34), "almost exclusively" sounds a bit too strong expression considering known several works of other luminescent radicals.

Response 5

Following the reviewer's comment, we have revised this sentence as follows:

“Majority of luminescent π -radicals are based on chlorinated triphenylmethyl (trityl) derivatives...”

Comment 6

In Page 9 (line 178), it is more reader-friendly to write photoluminescence quantum yield of TTM-3PCz for comparison.

Response 6

Following the reviewer's suggestion, we have added the photoluminescence quantum yield of TTM-3PCz as follows:

“We note that mesityl substitution also enhances the PLQE of D–A* type radical M₂TTM-3PCz from 46% (for TTM-3PCz) to 84% in toluene solution, which stems from the increase in radiative decay rate (Table 1 and Supplementary Fig. 34).¹⁶”

Comment 7

Fig. S5, PFTTM and PCzTTM would be typos of PFMTTM and PCzMTTM.

Response 7

We thank the reviewer for pointing this out. We have corrected the typos in the figure and its caption (Fig. S6 in the revised supplementary information file).

REVIEWERS' COMMENTS

Reviewer #1 (Remarks to the Author):

I am satisfied with the changes made by the authors and I now recommend the revised version of the manuscript for publication.

Reviewer #2 (Remarks to the Author):

The authors have performed sufficient experiments to clarify the ambiguities, especially on the type 2 band alignment between CBP and the titled compounds. After careful examination on the revised manuscript and supplementary information, I affirm that the current format is ready for publication on the journal Nature Communication, and further revisions are not required.

Reviewer #3 (Remarks to the Author):

The authors have performed additional experiments and have answered all questions and comments from the all reviewers.

The manuscript is recommended for publication.